# Impact of a Tripartite Collaboration between Oncologist, Pharmacist and Diabetologist in the Management of Patients with Diabetes Starting Chemotherapy: The ONCODIAB Trial

**DOI:** 10.3390/cancers15184544

**Published:** 2023-09-13

**Authors:** Justine Paris, Pauline Legris, Madeline Devaux, Stephanie Bost, Pauline Gueneau, Cedric Rossi, Sylvain Manfredi, Benjamin Bouillet, Jean-Michel Petit, Pauline Pistre, Mathieu Boulin

**Affiliations:** 1Department of Pharmacy, University Hospital, 21000 Dijon, France; justine.paris@chu-dijon.fr (J.P.); madeline.devaux@chu-dijon.fr (M.D.); stephanie.bost@chu-dijon.fr (S.B.); pauline.gueneau@chu-dijon.fr (P.G.); pauline.pistre@chu-dijon.fr (P.P.); 2Department of Endocrinology, Diabetes and Metabolic Disorders, University Hospital, 21000 Dijon, France; pauline.legris@chu-dijon.fr (P.L.); benjamin.bouillet@chu-dijon.fr (B.B.); jean-michel.petit@chu-dijon.fr (J.-M.P.); 3Department of Clinical Hematology, University Hospital and SAPHIIT UMR 1231, University of Burgundy & Franche Comte, 21000 Dijon, France; cedric.rossi@chu-dijon.fr; 4Department of Hepatogastroenterology and Digestive Oncology, University Hospital and EPICAD LNC UMR 1231, University of Burgundy & Franche Comte, 21000 Dijon, France; sylvain.manfredi@chu-dijon.fr; 5PADYS LNC UMR 1231, University of Burgundy & Franche Comte, 21000 Dijon, France; 6EPICAD LNC UMR 1231, University of Burgundy & Franche Comte, 21000 Dijon, France

**Keywords:** cancer, diabetes, diabetologist, pharmacist, medication optimization

## Abstract

**Simple Summary:**

Approximately 15% of cancer patients have diabetes. These patients often had difficulties in their glycemic control during chemotherapy periods. Patients suffering from these two diseases are often aged 65 years and older, with other cardiovascular comorbidities including renal failure and polypharmacy. Continuous glucose monitoring may help diabetologists in the glycemic management of these patients. We performed a study to evaluate a tripartite oncologist–pharmacist–diabetologist collaboration helped by continuous glucose monitoring records in patients with diabetes starting chemotherapy. A total of 106 consecutively recruited patients were included. Based on exploitable data for 94 patients, we demonstrated that the collaboration between oncologists, pharmacists, and diabetologists helped by continuous glucose monitoring led to overall medication optimization and better glycemic control at 6 months in patients with diabetes starting chemotherapy.

**Abstract:**

Background: Diabetes negatively impacts cancer prognosis. The objective of this work was to evaluate a tripartite oncologist–pharmacist–diabetologist collaboration in the management of patients with diabetes starting chemotherapy. Patients and Methods: The prospective ONCODIAB study (NCT04315857) included 102 adults with diabetes starting chemotherapy by whom a continuous glucose monitoring device was worn for fourteen days from the first day of the first and second chemotherapy cycles. The primary outcome was to assess pharmacist and diabetologist interventions. The secondary outcome was to evaluate the impact of the ONCODIAB follow-up on individualized patient glycemic targets at 6 months. Results: A total of 191 (2 per patient) were made either by clinical pharmacists (*n* = 95) or diabetologists (*n* = 96) during the first two chemotherapy cycles. The anatomic therapeutic chemical drug classes most frequently involved in pharmacist interventions were cardiovascular system (23%), alimentary tract and metabolism (22%), and anti-infectives for systemic use (14%). Diabetologists modified the antidiabetic treatment in 58 (62%) of patients: dose reduction (34%), drug discontinuation (28%), drug addition (24%), and dose increase (15%). Glycated hemoglobin decreased from 7.6 ± 1.7% at baseline to 7.1 ± 1.1% at 6 months (*p* = 0.02). Compared to individualized targets, HbA1c was higher, in the interval, or lower in 29%, 44%, and 27% of patients at baseline vs. in 8%, 70%, and 22% of patients at 6 months, respectively (*p* < 10^−3^). Conclusions: In our study, a close collaboration between oncologists, pharmacists, and diabetologists helped by continuous glucose monitoring led to overall medication optimization and better glycemic control in patients with diabetes starting chemotherapy.

## 1. Introduction

Diabetes and cancer are commonly coexisting illnesses and their global incidence and prevalence are rising [1]. Evidence suggests that cancer patients with diabetes have higher cancer-related mortality and morbidity [2,3].

The relationship between cancer, anticancer treatment, diabetes, and antidiabetic treatment is complex. Hypoglycemic episodes may be caused by nausea/vomiting, weight loss, anorexia, and malnutrition due to the cancer and/or to anticancer treatments including surgery and anticancer agents [4]. Conversely, anticancer agents as well as corticosteroids or iterative 5% dextrose infusions may lead to hyperglycemic episodes [4]. Diabetes-related chronic hyperglycemia during anticancer treatment increases the risk of infection and neuropathy that can lead to anticancer agent dose reduction, delay, or discontinuation [5,6,7]. The reciprocal negative impact of the two diseases and their associated treatments is maximized by the fact that patients suffering from the two diseases are often aged 65 years and older, with other cardiovascular comorbidities including renal failure and polypharmacy. In the management of patients with cancer, collaboration between oncologists and pharmacists is widely described [8]. In a single-center prospective study in cancer patients with diabetes, those randomized in the pharmacist intervention group showed a better glycemic control (*p* = 0.049), a significant increase in medication adherence (*p* = 0.0049), and a significant increase in diabetes self-care activities, including diet (*p* = 0.037), self-monitoring of blood glucose (*p* = 0.027) and foot care (*p* = 0.0085) at 3 months [9]. No continuous glucose monitoring (CGM) was used and no diabetologists were involved in the trial [9]. To our knowledge, no data have evaluated a multidisciplinary management of cancer patients with diabetes.

The aim of the present study was to evaluate a tripartite oncologist–pharmacist-diabetologist collaboration helped by CGM records in the management of patients with diabetes starting chemotherapy. The primary objective was to describe and assess the clinical impact of pharmacist and diabetologist interventions. The secondary objective was to evaluate the modification of glycated hemoglobin (HbA1c) 6 months after chemotherapy initiation.

## 2. Methods

### 2.1. UMACOACH Program

The UMACOACH (Unité Médicale Ambulatoire de Cancérologie cOllaboration Assistance Chimiothérapie) program was implemented in the ambulatory hematology–oncology department of our hospital in November 2017. It is a pharmacist-led program with the aim of optimizing drug management of patients with hematological malignancies and digestive, skin, and gynecological cancers treated by immuno- and/or chemotherapy. Other healthcare professionals are involved in the program to improve patient management: oncologists, clinical nurses, dietitians, and oncopsychologists.

In daily practice, clinical pharmacists carry out the pharmaceutical validation of immuno- and/or chemotherapy regimens using patient medical files and biological results (indication, dosage, and interval between cycles). A full medication review is performed at each treatment initiation based on patient interviews, analyses of prescriptions, clinical and biological data, and pharmacy dispensing. This best possible medication reconciliation enables the clinical pharmacists to detect any interactions between the patients’ usual treatment and that prescribed by the oncologist/hematologist (anticancer and adjuvant agents), leading to pharmacist interventions (PI). PI are solved or not on the day after contacting the different prescribers concerned (general practitioner or other physicians including oncologists/hematologists). Before discharge, all patients have a pharmaceutical consultation with the aim of maximizing patient knowledge (“what to do in case of…”). The new anticancer treatment is explained (mechanism for action, time for intake for oral agents, etc.) with the possible adverse events (AE) as well as adjuvant agents that are prescribed with their recommended use. In case of accepted PI, any reasons for changes in the usual treatment are explained at the end of the consultation. A pharmaceutical note including an exhaustive synthesis of all patient medicines, including vaccines, over-the-counter agents, and natural products, is sent by secured e-mail to the general practitioner, specialized physicians, community pharmacist, and nurse of the patient. More than 800 patients are enrolled in the UMACOACH program each year [8].

### 2.2. ONCODIAB Study

#### 2.2.1. Study Design

To optimize the management of diabetic patients starting chemotherapy for cancer, we specifically reinforced the UMACOACH program with an intensive glycemic follow-up through the ONCODIAB single-center, single-arm prospective study. Patients were recruited in the ambulatory hematology–oncology department of our teaching hospital. The study was performed in accordance with good clinical practice and the Declaration of Helsinki guidelines. The study was registered on ClinicalTrials.gov (identifier: NCT04315857). All patients provided oral informed consent before enrolment. Full details of the ONCODIAB study have been previously reported [10].

#### 2.2.2. Patients

Patients were eligible for inclusion if they (a) were older than 18 years, (b) started an anticancer treatment with at least one conventional anticancer agent, and (c) had diabetes with at least one antidiabetic drug. Patients were recruited consecutively on the day they started chemotherapy in the ambulatory hematology–oncology department. They were excluded if they had received an anticancer treatment within six months before inclusion.

#### 2.2.3. Interventions

Besides the UMACOACH program, the originality of the ONCODIAB study is to specifically reinforce the management of high-risk diabetic patients through early diabetologist visits helped by CGM records. During the first two chemotherapy cycles, all (single-arm) participants wore a sensor (Free Style Libre Pro; Abbott Diabetes Care, Alameda, CA, USA) measuring interstitial glucose levels in blinded mode (i.e., the participant and medical staff were unable to see the glucose values), over a 14-day period. The first sensor was applied on the initiation day of the first chemotherapy cycle. Based on the first CGM record and all data that might influence glycemic control collected during the visit at the time of the second cycle start (visit 1), the diabetologist possibly modified the antidiabetic treatment. Antidiabetic therapeutic changes implemented by the diabetologist were reexplained by the pharmacist before patient discharge. Before starting the third chemotherapy cycle, a second diabetologist visit was performed helped by the second CGM record with possible antidiabetic treatment changes. In case of longer-than-14-day intervals between the two chemotherapy cycles (scheduled or not), the second sensor was only applied on the initiation day of the second chemotherapy cycle. The ONCODIAB follow-up is presented in Figure 1.

#### 2.2.4. Outcomes and Statistical Analysis

The primary outcome of the present analysis was to describe and to assess pharmacist interventions (PI) as well as diabetologist therapeutic changes in the ONCODIAB trial. PI performed by clinician pharmacists is defined as “any action taken by a pharmacist that directly results in a change of patient management or therapy” [11]. We differentiated two categories of PI: PI performed with prescribers (PIpr) and PI performed with patients during pharmaceutical consultations (PI*pa*). A PI*pr* was considered as “ accepted“ if both the general practitioner or another physician who prescribed drug *x* (amiodarone, for example, is often prescribed by a cardiologist) and the hematologist/oncologist accepted the PI. In case of PI*pr* concerning the chemotherapeutic regimen or adjuvant drugs (antimicrobial agents, antiemetics, etc.), the acceptance was only required for the hematologist/oncologist. After PI*pr* acceptance, a systematic explanation of the therapeutic changes was given to the patient during the pharmaceutical consultation. A PI*pa* was considered as ‘accepted’ if the patient accepted it.

PI were classified according to the validated tool from the French Society of Clinical Pharmacy: identification of drug-related problem (eleven categories: contra-indication/non conformity to guidelines; dosage problem; drug interaction; adverse drug reaction; drug omission; drug or medical device not received; unjustified drug prescription; therapeutic redundancy; improper prescription; pharmacodependence; and monitoring), type of intervention (seven categories: dose adjustment; choice of the route administration; optimization of the dispensing/administration mode; drug monitoring; addition of a new drug; drug switch; and discontinuation or refusal to deliver), and the acceptance by the prescriber/patient [12]. The clinical impact of PI was rated according to seven levels of severity (lethal, harmful, major, moderate, minor, null, and non-determined) using the CLinical Economic and Organizational (CLEO) tool [13].

The secondary outcome was to evaluate the impact of the ONCODIAB follow-up on individualized patient glycemic targets at 6 months. For that, we first defined for each patient his/her target HbA1c level according to the 2019 French Diabetes Society guidelines [14]. In the guidelines, the target may vary between ≤6.5% in newly diagnosed younger adults and ≤9% in frail older adults. We compared this theoretical target level to that of the patient before starting chemotherapy (baseline HbA1c) and that obtained six months after chemotherapy initiation (6-month HbA1c). We also described the antidiabetic treatment at 6 months. The HbA1c levels as well as the antidiabetic treatment were obtained by phone calls to laboratories, patients, and/or community pharmacies.

In the absence of any data in the literature on therapeutic changes made by a multidisciplinary team in patients with diabetes starting chemotherapy, we arbitrarily defined a final sample size of 100 patients including a potential drop-out rate of 5%. Qualitative and quantitative variables were described using frequencies and percentages or means ± standard deviations. To compare baseline and 6-month HbA1c, chi-square or Fisher’s Exact Test were performed. Statistical significance was set at *p* ≤ 0.05. All analyses were performed in Microsoft^®^ Excel Version 16.11.

## 3. Results

### 3.1. Patient Baseline Characteristics

Between 1 June 2020 and 1 March 2022, 106 patients were consecutively screened for the ONCODIAB study. Four (4%) patients refused the study; eight (8%) were not included in the final analysis because of technical problems with the FSL sensor captor and/or diabetologists’ inability to interpret data. The baseline characteristics of the ninety-four patients included in the analysis are summarized in Table 1.

Patients were treated for digestive, hematological, or gynecological cancer in 57%, 30%, and 13% of the cases, respectively. Diabetes treatment was a monotherapy (48%), a bitherapy (26%), or a tritherapy or more (26%). The most prescribed antidiabetic agents were metformin (61%), long-acting insulins (38%), rapid-acting insulins (28%), dipeptidyl-peptidase-4 (DPP-4) inhibitors (20%), repaglinide (19%), sulfonylureas (18%), and glucagon-like peptide-1 (GLP1) receptor agonists (15%).

### 3.2. Pharmacist and Diabetologist Interventions

In the 94 patients with two consecutive CGM recordings, 191 therapeutic changes (2.0 per patient) were made either after PI or by the diabetologist during visit 1 or visit 2.

Ninety-five PI were performed, corresponding to a mean of 1.0 PI per patient. There were 87% PIpr and 13% PIpa. The clinical impact was classified as major, moderate or minor in 19%, 67%, and 14% of cases, respectively. The anatomic therapeutic chemical drug classes most frequently involved in PI were cardiovascular system (23%), alimentary tract and metabolism (22%), antiinfectives for systemic use (14%), nervous system (10%), various (10%), and antineoplastic and immunomodulating agents (8%). All drug-related problems, PI, and clinical impacts are detailed in Table 2.

A total of 96 antidiabetic treatment modifications were performed by diabetologists in 58 (62%) patients corresponding to a mean of 1.0 (96/94) per patient. There were no antidiabetic treatment changes in 36 (38%) patients. In the other 58 patients, the modifications were dose reductions (34%), drug discontinuations (28%), drug additions (24%), and dose increases (15%). Drug discontinuations concerned sulfonylureas (*n* = 7, 25%), GLP1 receptor agonists (*n* = 5, 18%), insulin (*n* = 4, 14%) metformin (*n* = 4, 14%), repaglinide (*n* = 2, 7%), DPP-4 inhibitors (*n* = 2, 7%), and acarbose (*n* = 1, 4%). Finally, 6 (6%) patients had no more antidiabetic treatments. An antidiabetic class switch was performed in 9 (9%) patients, for example, from repaglinide to DPP-4 inhibitor or sulfonylurea plus DPP-4 inhibitor to metformin in case of hypoglycemia, and from GLP1 receptor agonist to long-acting insulin plus glinide in case of hyperglycemia. The most prescribed agents after diabetologist visits were metformin (56%), long-acting insulin (39%), rapid-acting insulin (28%), glinide (24%), and DPP-4 inhibitor (20%) (Figure 2).

### 3.3. Glycemic Target and Antidiabetic Treatment at 6 Months

HbA1c at 6 months was collected for 87 patients (six cancer-related deaths, one HbA1c test not performed). The mean baseline and 6-month HbA1c were 7.6 ± 1.7% and 7.1 ± 1.1% (*p* = 0.02). According to guidelines, the individualized target HbA1c was estimated to be ≤6.5% for 1 patient, ≤7% for 19 patients, ≤8% for 58 patients (but ≥7% for those treated with insulin, sulfonylurea, and/or repaglinide; otherwise, they were considered as below the target), and ≤9% for 9 patients (but ≥7% for those treated with insulin, sulfonylurea, and/or repaglinide). Compared to the target, the baseline HbA1c was higher, in the interval, or lower in 29%, 44%, and 27% of patients, respectively. The most frequently prescribed agents in patients with a baseline HbA1c below the target were metformin (46%), repaglinide (33%), long-acting insulins (25%), DPP-4 inhibitors (25%), and sulfonylureas (25%). Compared to the target, the 6-month HbA1c was higher, in the interval, or lower in 8%, 70%, and 22%, respectively (Figure 3).

A significant difference was observed between baseline and 6-month frequencies of patients above, in the interval, and below the HbA1c target (*p* < 10^−3^). The most frequently prescribed agents in patients with a 6-month HbA1c below the target were metformin (53%), repaglinide (26%), long-acting insulin (21%), sulfonylurea (16%), and DPP-4 inhibitor (16%).

The antidiabetic treatment at 6 months was collected for 88 patients (six deaths). For 63 (72%) patients, antidiabetic treatment did not change since the second diabetologist visit; 4 (4%) patients were prescribed their baseline pre-chemotherapy antidiabetic treatment after a general practitioner visit; and 21 (24%) patients had antidiabetic treatment changes. Modifications were made by the patient’s general practitioner or the diabetologist. There were dose increases (*n* = 9), dose decreases (*n* = 5), drug switches (*n* = 3, one from metformin to repaglinide and two from DPP-4 inhibitor to insulin), drug additions (*n* = 2, one for insulin in a patient after cephalic duodenopancreatectomy resection, and one for metformin), and antidiabetic treatment discontinuation (*n* = 2, metformin and DPP-4 as single agent).

## 4. Discussion

We reported two therapeutic modifications/patient in the ONCODIAB follow-up. That represents an elevated number compared to the literature [8,15,16]. Several reasons may explain this result. First, the originality of the UMACOACH ONCODIAB management is to focus on diabetes. The probability of therapeutic changes after two Freestyle Libre Pro^®^ recordings and two scheduled diabetologist visits was intrinsically high. This is confirmed by the elevated ninety-six antidiabetic treatment modifications only performed by the diabetologist. To our knowledge, our study is the first to include an intensive glycemic follow-up at chemotherapy initiation with diabetologist interventions helped by CGM during the first two chemotherapy cycles. Second, the collaboration of a diabetologist with a clinical pharmacist is efficient, with the diabetologist focusing his/her action on antidiabetic treatment while the pharmacist focused his/her PI on the overall treatment except antidiabetic drugs. Third, this collaboration alongside oncologists/hematologists is probably feasible and efficient because patients with diabetes and cancer are patients with an elevated incidence of drug-related complications. They are often old with other comorbidities and an elevated number of medicines [17].

The more significant result of the ONCODIAB follow-up is the significant improvement of glycemic control at 6 months. Even if widely questionable, HbA1c remains the gold standard to evaluate the glycemic control of a patient. We chose to evaluate HbA1c at 6 months to evaluate the midterm impact of the specific diabetologist interventions during the first two chemotherapy cycles. Clinical studies in cancer patients with diabetes evaluated the proportion of those reaching a fixed HbA1c level or the time to reach a HbA1c level below 7% [1,9]. We decided in the present study to evaluate the proportion of patients reaching their individualized targets as recommended by more recent international guidelines and knowing the potentially high proportion of older frail patients starting an anticancer treatment. The proportion of patients with HbA1c levels above the target decreased by 73% (from 29% to 8%), decreasing chronic hyperglycemia and its associated infectious complications maximized during neutro- and lymphopenic periods between chemotherapy cycles [1,18]. Long-term decreasing exposure to hyperglycemia also reduces diabetes-related micro- and macrovascular complications in particular in patients who will survive their cancer [19,20]. HbA1c levels below the target were reduced by 19% (from 27% to 22%).

The difficulty in normalizing HbA1c below the target could be explained by factors inducing hypoglycemia independently of antidiabetic agents, such as nausea and vomiting, undernourishment, and anorexia due to cancer and/or anticancer treatments. Another explanation of the slight decrease in HbA1c below the target is the elevated use of hypoglycemic agents at 6 months: repaglinide (26%), long-acting insulin (21%), and sulfonylureas (16%). Moreover, in the last fifteen years, a number of studies have described a potential overtreatment in older adults with diabetes and its deleterious impact on patients [21,22,23]. We thus hypothesized that the major clinical challenge when starting chemotherapy in diabetic patients would be to avoid hypoglycemic episodes and their potentially severe complications. No severe hypoglycemic episodes occurred in this study. They might have been reduced by diabetologist medication optimization and the educational messages delivered by both diabetologist and pharmacist.

Our study has several strengths. To our knowledge, it is the first prospective study integrating an intensive glycemic follow-up in cancer patients with diabetes starting chemotherapy. No specific criteria were applied except that patients should start a chemotherapy regimen including at least one conventional agent. All conventional agents that are still widely used in clinical practice are characterized by emetic and anorexia properties [24]. No patients receiving immunotherapy or targeted oral/injectable therapies except if associated with a conventional agent were included. All patients were included consecutively with a very low (4%) rate of refusal. This probably illustrates preoccupations the patients might have concerning self-managing their diabetes during chemotherapy cycles [25]. For diabetologists, the use of a continuous interstitial blood glucose sensor accurately highlighted periods with hypo- and/or hyperglycemia, helping them to adjust the antidiabetic medication. Finally, no patients reported captor-related adverse events.

There are some limitations in this study. First, it was a monocentric study. Second, our population was heterogeneous with different types of cancer and chemotherapeutic regimens as well as different diabetes and antidiabetic medication profiles. However, no selection was carried out when consecutively including approximately 100 patients. Thus, they are probably representative of diabetic patients with hematological, digestive, and gynecological cancers in France. Third, we did not take into account the factors that could have influenced the variations of HbA1c over the 6-month follow-up period.

## 5. Conclusions

A close collaboration between oncologists, pharmacists, and diabetologists helped by continuous glucose monitoring led to overall medication optimization and better glycemic control in patients with diabetes starting chemotherapy.

## Figures and Tables

**Figure 1 cancers-15-04544-f001:**
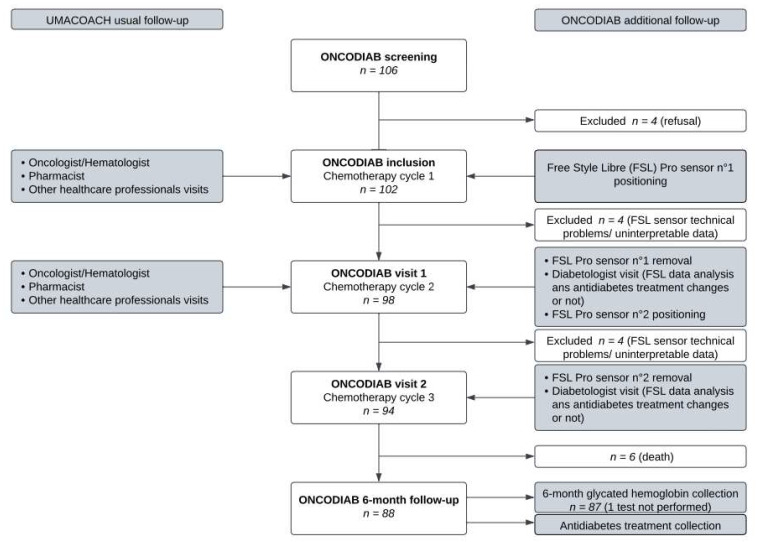
Description of UMACOACH ONCODIAB follow-up. The intensive glycemic ONCODIAB follow-up is presented on the right of the figure.

**Figure 2 cancers-15-04544-f002:**
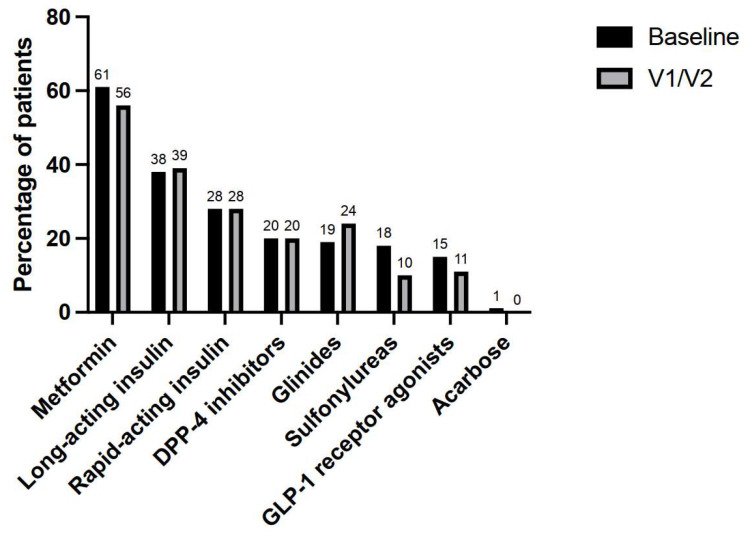
Percentages of patients receiving each one of the antidiabetic medication classes before and after first (V1) and second (V2) diabetologist interventions.

**Figure 3 cancers-15-04544-f003:**
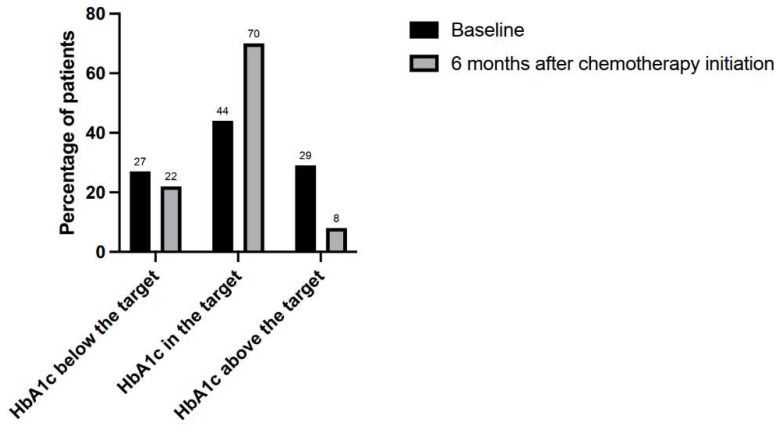
Percentages of patients below, in the interval, and above their individualized HbA1c targets at baseline and 6 months after chemotherapy initiation.

**Table 1 cancers-15-04544-t001:** Patient baseline characteristics.

Characteristics	*N* = 94
**Age (years) Mean ± SD**	69.1 ± 8.9
**Weight (kg) Mean ± SD**	79.2 ± 18.4
**Body Mass (kg/m^2^) ± SD**	27.3 ± 6.2
**Gender, *n* (%)** Male Female	55 (59) 39 (41)
**Diabetes Type, *n* (%)** Type 1 Type 2 and other	2 (2) 92 (98)
**Diagnosis of Diabetes, (years) *n* (%)** 0–1 2–10 >10	11 (12) 41 (44) 42 (45)
**Antidiabetic Drug Class, *n* (%)** Metformin Long-acting insulin Rapid-acting insulin DPP-4 inhibitors Glinides Sulfonylureas GLP-1 receptor agonists Acarbose	57 (61) 36 (38) 26 (28) 19 (20) 18 (19) 17 (18) 14 (15) 1 (1)
**Antidiabetic Treatment, *n* (%)**	
Monotherapy	45 (48)
Bitherapy	24 (26)
Tritherapy or More	25 (26)
**Glycated hemoglobin (%) mean ± SD**	7.56 ± 1.74
**Cancer Type *N* (%)** Digestive Hematologic Gynecologic	54 (57) 28 (30) 12 (13)

**Table 2 cancers-15-04544-t002:** Description of pharmacist interventions (PI).

	Drug-Related Problem	*N* (%)	PI	*N* (%)
**PI with Prescribers** ***n* = 83 (87%)**	Contra-indication/non-conformity to guidelines	22 (27)	Addition of a new drug	30 (36)
	Dosage problem (under or over dosage)	12 (14)	Discontinuation or refusal to deliver	18 (22)
	Drug or medical device not received by the patient	12 (14)	Dose adjustment	16 (19)
	Drug omission	9 (11)	Drug monitoring	9 (11)
	Monitoring	9 (11)	Drug switch	7 (8)
	Improper prescription	6 (7)	Optimization of the dispensing/administration mode	3 (4)
	Unjustified drug prescription	5 (6)		
	Adverse drug reaction	4 (5)		
	Drug interaction	3 (4)		
	Therapeutic redundancy	1 (1)		
**PI with Patients** ***n* = 12 (13%)**	Improper prescription	12 (100)	Optimization of the dispensing/administration mode	12 (100)
**Clinical Impact of PI** ***n* = 95**	Moderate	64 (67)		
	Major	18 (19)		
	Minor	13 (14)		

## Data Availability

The data presented in this study are available on request from the corresponding author.

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
