# Peer review of "Impact of a Tripartite Collaboration between Oncologist, Pharmacist and Diabetologist in the Management of Patients with Diabetes Starting Chemotherapy: The ONCODIAB Trial"

_cancers, 2023, doi:10.3390/cancers15184544_

Round 1

Reviewer 1 Report

The paper has documented important information regarding a scientific study of collaborative care management of children's diabetes. Although the sample size is relatively small, the prospective study design enables to tease out the positive benefits or effects of collaborative care for monitoring diabetes care and outcomes. The detailed presentation of the collaborative care arrangement for diabetes is helpful for the improvement of continuous care for a chronic condition. The only minor amendment needed is as follows:  Is the severity of disease being considered in the prospective study? 

Reviewer 2 Report

• The "Introduction" part of the study should be expanded, considering the research objectives, problems, and hypotheses.  • What are the inclusion and exclusion criteria in the study? • The design of the study should be specified in the Materials and Methods section.  • Research guideline(s)/standard(s) appropriate to the study design should be reported in the paper text. • Which randomization method was used in the distribution of the individuals included in the study to the groups?  • Which blinding (masking) method was used in the study?  • The primary output/endpoint variable(s)/measurement(s) of the study should be defined.  • How was the sample size determined? This information should be explained in the Materials and Methods section.  • Which sampling (probable or non-probable, etc.) method was used in the study?  • Statistical tests for hypothesis testing and their assumptions should be specified in the study's statistical analysis in the Materials and Methods section.  • The details (version, license number, etc.) of the statistical package(s) or program(s) should be given in the section of "Data Analysis or Statistical Analysis". • It should be explained how the qualitative and quantitative data are summarized under the sub-heading of Statistical Analyses in the Materials and Methods section of the study.  • The exact P values should be added to the table(s) (e.g., p=0.25; p=0.03).  • Which methods are used to model relationships between variables?  • The descriptions and other descriptive values/data should be defined on the tables and shapes. • Are the data subjected to pre-processing?  • How were extreme/outlier values in the data determined and resolved?

Reviewer 3 Report

Overall, this research article is interesting because it involves collaboration between oncologists, pharmacists, and diabetologists in managing patients with diabetes starting chemotherapy. The explanation of the methods and results is quite detailed. In the display of the results of table sections 1 and 2, you should tidy up again and use a font size smaller than the font size in the body of the article. Figure. 2 should be made with a more attractive appearance with certain software, not only using Excel. You should add references related to other research in the discussion section in each paragraph.

Reviewer 4 Report

Paris et al reported a clinical trial to test the effects of collaboration between oncologists, pharmacist and diabetologist on the management of cancer patients with diabetes. The design and results of this trial is adequately described but some improvement can be made before the publication.
1.
Figure 1 should include numbers of total patients enrolled in the beginning, dropped or
excluded at each stage and reasons, and size of population for final analysis.

2.
Figure 2 is not clear described. I assume the the number above the bars are patient numbers, not percentages. Then the number of sulfonylureas decreased from 18 to 10. However, only 7 patients discontinued this drug in the text. This is confusing.

3.
The description on HbA1c seems over simplified. A graph of paired comparison can be draw, I.e. individual change of HbA1c levels. It would be greatly helpful to show if those
levels were interval at baseline are still in the interval after 6 months, and if more people had normalized HbA1c than people got deteriorated.

4.
Do authors have any reference data on the management of HbA1c levels in the absence of such a collaboration? Does the collaboration do better than, or the same as without
it? Authors may want to discuss or estimate it.

5.
Has authors measured other metrics like blood glucose level?

6.
In regard to the inclusion criteria, was the patient planned to start the treatment after enrollment? Please make it clear.

7.
Please proofread the whole manuscript to remove typing errors. For example,  "betes" and  "e prospective" in the abstract.

8.
The abstract mentioned that 191 therapeutic changes were collected during two
chemotherapy cycles. However Figure 1 indicates three cycles. Please confirm whether
they were all in the first two rounds (baseline, visit 1) or third r
ound (visit 2).

Paris et al reported a clinical trial to test the effects of collaboration between oncologists, pharmacist and diabetologist on the management of cancer patients with diabetes. The design and results of this trial is adequately described but some improvement can be made before the publication.
1.
Figure 1 should include numbers of total patients enrolled in the beginning, dropped or
excluded at each stage and reasons, and size of population for final analysis.

2.
Figure 2 is not clear described. I assume the the number above the bars are patient numbers, not percentages. Then the number of sulfonylureas decreased from 18 to 10. However, only 7 patients discontinued this drug in the text. This is confusing.

3.
The description on HbA1c seems over simplified. A graph of paired comparison can be draw, I.e. individual change of HbA1c levels. It would be greatly helpful to show if those
levels were interval at baseline are still in the interval after 6 months, and if more people had normalized HbA1c than people got deteriorated.

4.
Do authors have any reference data on the management of HbA1c levels in the absence of such a collaboration? Does the collaboration do better than, or the same as without
it? Authors may want to discuss or estimate it.

5.
Has authors measured other metrics like blood glucose level?

6.
In regard to the inclusion criteria, was the patient planned to start the treatment after enrollment? Please make it clear.

7.
Please proofread the whole manuscript to remove typing errors. For example,  "betes" and  "e prospective" in the abstract.

8.
The abstract mentioned that 191 therapeutic changes were collected during two
chemotherapy cycles. However Figure 1 indicates three cycles. Please confirm whether
they were all in the first two rounds (baseline, visit 1) or third r
ound (visit 2).

Reviewer 5 Report

In this article, the authors study the impact of a tripartite collaboration between oncologist, pharmacist and diabetologist in the management of patients with diabetes starting chemotherapy for different tumor types.

General comment:

The results of the study are relevant. The limitations of the study are fairly presented.

Minor point:

Figure 1: The quality of the Figure should be enhanced (it is a bit blurred).

Minor editing of English language required.

Round 2

Reviewer 1 Report

The revision is acceptable.  No further comments are given.

The paper could be recommended for further consideration for publication.

Reviewer 3 Report

The author has revised the article according to the suggestion that has been given.

Reviewer 4 Report

The authors have addressed all my points and I'd like to recommend acceptance of this manuscript.

Reviewer 5 Report

The manuscript is improved now.